# The Cellular Structure and Toughness of Hydrogenated Styrene-Butadiene Block Copolymer Reinforced Polypropylene Foams

**DOI:** 10.3390/polym15061503

**Published:** 2023-03-17

**Authors:** Wei Guo, Zicheng Zheng, Wei Li, Hao Li, Fankun Zeng, Huajie Mao

**Affiliations:** 1Hubei Key Laboratory of Advanced Technology for Automotive Components, Wuhan University of Technology, Wuhan 430070, China; 2Hubei Collaborative Innovation Centre for Automotive Components Technology, Wuhan University of Technology, Wuhan 430070, China; 3Hubei Research Center for New Energy & Intelligent Connected Vehicle, Wuhan University of Technology, Wuhan 430070, China; 4Institute of Advanced Materials and Manufacturing Technology, Wuhan University of Technology, Wuhan 430070, China; 5SAIC-GM-Wuling Automobile Co., Ltd., Liuzhou 545007, China; 6School of Materials Science and Engineering, Wuhan University of Technology, Wuhan 430070, China

**Keywords:** microcellular polypropylene, thermoplastic elastomer, SEBS, toughening

## Abstract

Polypropylene nanocomposites containing varying amounts of Styrene-ethylene-butadiene-styrene block copolymer (SEBS) were prepared through the supercritical nitrogen microcellular injection-molding process. Maleic anhydride (MAH)-grafted polypropylene (PP-g-MAH) copolymers were used as compatibilizers. The influence of SEBS content on the cell structure and toughness of the SEBS/PP composites was investigated. Upon the addition of SEBS, the differential scanning calorimeter tests revealed that the grain size of the composites decreased, and their toughness increased. The results of the rheological behavior tests showed that the melt viscosity of the composite increased, playing a role in enhancing the cell structure. With the addition of 20 wt% SEBS, the cell diameter decreased from 157 to 66.7 μm, leading to an improvement in the mechanical properties. Compared to pure PP material, the impact toughness of the composites rose by 410% with 20 wt% of SEBS. Microstructure images of the impact section displayed evident plastic deformation, effectively absorbing energy and improving the material’s toughness. Furthermore, the composites exhibited a significant increase in toughness in the tensile test, with the foamed material’s elongation at break being 960% higher than that of pure PP foamed material when the SEBS content was 20%.

## 1. Introduction

Microcellular foamed polypropylene (PP) materials are commonly used in automobiles due to their low density, high wave absorption, and good thermal and mechanical properties. However, their low melt strength, poor impact resistance, narrow foaming temperature range, and the tendency for gas escape during foaming, along with severe cell merging and collapse, limit their potential for further use [1]. To overcome these limitations, researchers have studied the reinforcement mechanisms of pure PP [2,3,4]. Previous studies have demonstrated that adding elastomers can effectively improve the toughness of polypropylene and its cell structure. For example, Liu et al. [4] used polyolefin elastomers (POE) and high-density polyethylene (HDPE) to modify polypropylene in blends, resulting in a 13-fold increase in impact strength compared to pure PP. Similarly, Das et al. [5] prepared PP-cp/SBS blends by adding SBS to the PP-cp matrix, and the impact test results showed that the impact performance of the blends was 12 times higher than that of pure PP-cp at −40 °C. Mao et al. [6] added POE particles to PP/CaCO_3_ blends, achieving the best foaming effect and highest impact strength while also improving tensile and flexural strengths with 10% POE added. Parameswaran Pillai et al. [7] prepared a ternary blend based on polypropylene (PP)/high-density polyethylene (HDPE)/ethylene propylene diene monomer (EPDM) and its composite with graphene nanosheets. Ma et al. [8] prepared a series of dynamically vulcanized isotactic polypropylene (PP)/ethylene-propylene-diene rubber thermoplastic vulcanized elastomeric vulcanizates (TPV) containing different levels of β-nucleating agent (β-NA) and introduced them into the isotactic PP matrix as toughening agents. The impact strength of the toughened blends was ten times higher than that of pure PP when the β-NA content was 0.5 wt%.

Styrene-ethylene-butadiene-styrene block copolymer (SEBS) is produced through the hydrogenation of styrene-butadiene-styrene (SBS). It boasts excellent mechanical properties and resistance to aging and corrosion [9]. SEBS is a formidable toughening agent, with its block structure possessing a higher melting point in the hard segment than in the soft segment. This higher melting point allows the hard segment to serve as a heterogeneous nucleation site, thus improving the cell structure. Meanwhile, the soft segment is more flexible than the hard segment and can contribute to the toughening effect. Studies by Gupta et al. [10,11,12] have shown that blending PP with SEBS can effectively enhance processing and impact resistance. Blending 15% SEBS with PP was found to increase the impact strength of PP by a factor of six. Similarly, Bassani et al. [13] found that SEBS can effectively increase the toughness of PP, with the impact strength of blends containing 20 wt% SEBS reaching up to 25 times that of pure PP. Rishi Sharma et al. [14] prepared PP/SEBS-g-MAH blends and increased the toughness of the blends by increasing the concentration of SEBS-g-MAH and the amorphization of PP. This resulted in increased molecular chain deformation of PP, which lead to a higher tensile modulus and an increase in elongation at break. The tensile and fracture toughness of the composites were also increased, and the grain size was reduced. Tjong et al. [15] blended SEBS-g-MAH with PP/organically modified montmorillonite (Org-MMT) and observed increased tensile and fracture toughness as well as a reduction in grain size. Marco Monti [16] utilized SEBS and olefin block copolymer (OBC) to modify the blend of recycled polypropylene and talc, significantly improving the impact properties of the prepared composites. Although previous studies have demonstrated the effectiveness of SEBS in toughening polypropylene, fewer studies have explored the impact of SEBS and cells on composite toughness.

In this study, microcellular foamed PP/SEBS composites with different SEBS contents were prepared using the microcellular injection molding (MIM) technique. The microstructure morphology of the composites was observed, and their mechanical properties were tested to investigate the effect of SEBS on toughness and foaming.

## 2. Materials and Methods

### 2.1. Materials

The polypropylene pellets utilized in this study were produced by Sinopec Yanshan Petrochemical Co., Ltd. (Beijing, China), with a brand name of K8303 and a density of 0.9 g/cm^3^. The pellets also have a melt flow rate (MFR) of 3 g/10 min. The MFR test temperature was set at 230 °C and the weight mass was 2.16 kg. The Styrene-ethylene-butadiene-styrene block copolymer (SEBS) used in this study was produced by Corten, with a brand name of G1650 and a density of 0.91 g/cm^3^, as well as a melt flow rate of 1 g/10 min. The MFR test temperature was set at 230 °C and the weight mass was 2.16 kg. The Maleic anhydride grafted polypropylene (PP-g-MAH) was produced by Dong yuan Ziheng Plastics Co., Ltd. (Dongguan, China), with a grafting rate of 1.2%. Nitrogen (N_2_), with a purity of 99%, was used as the blowing agent and was supplied by Wuhan Xiangyun Industrial Co., Ltd., Wuhan, China.

### 2.2. Composite Material Preparation

In this experiment, five sets of samples with different contents were prepared (Table 1). The first group was a control group that contained a compatibilizer. The PP matrix blended with the compatibilizer was referred to as MPP. The raw materials were blended proportionally using a mixer (SHR-10, Wuhan Yi Yang Plastic Machinery Co., Ltd., Wuhan, China), and then extruded and pelletized using a twin-screw extruder (SHJ-20, Nan-jing Giant Machinery Co., Ltd. (Nanjing, China)). The extruder was operated at a temperature of 185, 190, 195 and 195 °C from the barrel to the die. The main screw rotated at 180 rpm, and the feeder rotated at 160 rpm. The composite masterbatch was dried in a vacuum drying oven (101A-1, Shanghai Guangdi Instruments Co., Ltd. (Shanghai, China)) at 80 °C for 6 h, and then microcellular foam injection parts were manufactured using an injection molding machine (HDX50, Ningbo Haida Plastic Machinery Co., Ltd. (Ningbo, China)). The injection-molding machine had a temperature range of 190 to 200 °C, an injection pressure of 80 MPa, a gas saturation pressure of 16 MPa, and a cooling time of 16 s. The microcellular injection molding (MIM) technique process flow is shown in Figure 1.

### 2.3. Characterization

#### 2.3.1. Crystallization Behavior Test

The non-isothermal crystallization behavior of MPP and MPP/SEBS composites was investigated using a differential scanning calorimeter (DSC214, NETZSCH Instruments Manufacturing GmbH, Selb, Germany). For the DSC test, foamed samples weighing approximately 10 mg were ramped up from 30 to 230 °C at a ramp rate of 20 °C/min under a nitrogen atmosphere. The samples were held at a constant temperature of 230 °C for 5 min after completion and then cooled down to 30 °C at a cooling rate of 10 °C/min to eliminate the thermal history of the material. This procedure was repeated to complete the test. The DSC data obtained were analyzed using the NETZSCH Proteus-70 software. The following equation was used to estimate the crystallinity:(1)$$X_{C}=\frac{\Delta H_{m}}{\Delta H_{0}\cdot \phi }\times 100\%$$
where $\Delta H_{m}$ is the enthalpy of fusion of PP in the blend, $\Delta H_{0}$ is the enthalpy of PP with 100% crystallinity, assumed to be 209 J/g [17], and ϕ is the weight fraction of PP in the blend.

#### 2.3.2. Rheological Behavior Test

The rheological properties of MPP and PP/SEBS composites were evaluated using a capillary rheometer (CR-6000, High-Tech Testing Instruments (Dong guan) Co., Ltd. (Dongguan, China)). The pre-dried pellets were heated in a cylinder at 230 °C, and corresponding material shear rate and viscosity values were obtained for seven incremental drop rates. Three sets of rheological properties were measured for each component of the material.

#### 2.3.3. Characterization of Cell Structure

A scanning electron microscope (SEM, JSM-IT300, Nippon Electron Co., Ltd. (Tokyo, Japan)) was used to observe the microstructure morphology of both solid and foamed samples. The sampling locations are indicated in Figure 2. To prepare the samples for SEM analysis, they were immersed in liquid nitrogen for 1 h to undergo cryogenic fracture treatment, and the surface of the cross-section was sprayed with platinum to enhance the conductivity of the cross-section. The cell density, average cell diameter, and foaming multiplicity of the composite were analyzed using Image-Pro software. At least 200 cells were counted in each SEM image. The densities of the solid ($\rho_{0}$) and foamed ($\rho_{f}$) samples were measured using thermogravimetric analysis (TGA). The remaining parameters were calculated separately using the following equations [18,19]:(2)$$N_{0}=\delta {\left(\frac{n}{A}\right)}^{3/2}$$
(3)$$\bar{D}=\frac{1}{n}\sum_{i}^{n}D_{i}$$
(4)$$\delta =\frac{\rho_{s}}{\rho_{f}}$$
(5)$$\varepsilon =1-\frac{\rho_{f}}{\rho_{s}}$$
where $N_{0}$ is the cell density of PP foams (units/cm^3^), δ is the expansion ratio, *n* is the number of cells in the selected SEM image, A is the area of the SEM image (cm^2^), $\bar{D}$ is the average diameter of cells, $\rho_{f}$ is the density of the foamed sample (g/cm^3^), and $\rho_{s}$ is the density of the solid sample (g/cm^3^),ε represents the void fraction.

A polarized optical microscope (POM) was used to observe the spherulite morphology of the composites. The samples were prepared by heating the pre-dried pellets to 230 °C at a temperature increase rate of 20 °C/min, holding at a constant temperature for 5 min, and then lowering to room temperature at a temperature decrease rate of 10 °C/min. The spherulite morphology of the composites was observed by a polarized optical microscope (POM) (Carl-Zeiss/Axio Scope Al A, Oberkochen, Germany).

#### 2.3.4. Mechanical Performance Test

An electronic universal testing machine (CMT6104, Meister Industrial Systems (China) Co., Ltd. (Shanghai, China)) was used for tensile testing. The impact strength of the material was tested using a cantilever impact tester (XJUD-5.5, Chengde Jinjian Testing Instruments Co., Ltd. (Chengde, China)). The impact test was conducted according to GB/T 1843-2008 with an impact energy of 2.75 J. All experiments were conducted at room temperature, and the average value of ten tests was taken for each sample to reduce error.

The tensile properties of the materials were characterized using an electronic universal testing machine (CMT6104, Meister Industrial Systems (China) Co., Ltd. (Shanghai, China)). Meanwhile, the impact strength was evaluated using a cantilever impact tester (XJUD-5.5, Chengde Jinjian Testing Instruments Co., Ltd.) based on GB/T 1843-2008 standard with an impact energy of 2.75 J. All tests were performed at room temperature, and the average value of ten samples was reported to minimize experimental error.

## 3. Results and Discussion

### 3.1. Crystallization Behavior

Figure 3 shows the non-isothermal crystallization behavior of the MPP/SEBS blends. The crystallization results are listed in Table 2. The crystallization temperature (Tc) in the MPP/5%SEBS samples shifts to higher temperatures and gradually decreases with the increase of SEBS content. This is typically attributed to a decrease in the system’s crystallinity, which is in line with the data in Table 2. Meanwhile, the crystallinity of the blends slightly increases and then gradually decreases as the SEBS content increases (Table 2). This is because SEBS, as an elastomer, contains regions that are compatible with the PP matrix and act as nucleation sites, promoting crystallization and leading to an increase in crystallinity. However, when the content of SEBS continues to rise, it becomes an amorphous polymer and starts to aggregate, hindering the movement of PP molecular chains and reducing the crystallinity. This decrease in crystallinity is reflected in a decrease in the tensile as seen in the mechanical performance test. On the other hand, the elastomeric properties of SEBS increase the material’s toughness and elongation at break.

The DSC crystallization analysis can provide a quantitative characterization of the enthalpy shift of the system. However, the calculated crystallinity values only characterize the system’s crystallization ability singularly and do not reflect any trends in the crystal size of the system with increasing toughening particle content. Since the crystal structure plays a crucial role in the properties of a material, it is important to study and characterize it. This can be done using a polarized optical microscope (POM), which uses polarized light to determine if a substance is isotropic or anisotropic. In this experiment, POM was used to observe the crystal morphology of the composite material. 

Figure 4 shows the spherulite morphology of the hybrid system characterized by POM. The blue area in the figure represents the MPP crystalline region, and the yellow SEBS crystalline regions are dispersed throughout the MPP crystalline tissue (as indicated by the dashed box). As shown in Figure 4a, MPP exhibits more prominent grains, clearer grain boundaries, and fewer crystalline regions. However, as the SEBS content increases, the number of spherulites also increases, while the size of the spherulites decreases and the grain boundaries become blurred. This is because the SEBS acts as a heterogeneous nucleator, increasing the number of spherulites. At the same time, the presence of SEBS in the hybrid system hinders the movement of PP chain segments, which makes it more difficult for them to enter the crystallization unit and inhibits grain growth. This results in smaller spherulites, which positively impact the toughness of the blends. These observations are consistent with the findings of Denac et al. [20,21] and are reflected in the mechanical properties of the material.

### 3.2. Rheological Behavior

Rheological properties are important material properties that reflect the viscosity of a material. The viscosity of the composite is an indicator of the resistance encountered by the cell nucleation and growth during the foaming process, and it has a significant impact on the resulting cell structure and the mechanical properties of the material. 

The shear viscosity curves of the composites are depicted in Figure 5. Figure 5a shows that the increase in shear rate resulted in a decrease in the viscosity of the composites, which facilitated the homogeneous mixing of the gas and melt. On the other hand, Figure 5b demonstrates that 20% SEBS content at a low shear rate increased the viscosity of the composites from 830.4 to 1182.4 Pa·s. This is because SEBS has a higher viscosity than PP [22]. Therefore, the addition of SEBS results in a higher viscosity of the composites. Moreover, Figure 5b also illustrates that increasing the SEBS content leads to a steeper slope of the viscosity curve at low shear rates, meaning that the addition of SEBS makes the composites more sensitive to shear rates. As depicted in Figure 5c, the viscosity at 10% SEBS content is comparable to that at 20% SEBS content. This is because PP and SEBS are not entirely compatible, and a small amount of SEBS can effectively alter the physical cross-linked structure of the composite melt, thus improving its flowability. However, excessive SEBS increases the viscous resistance to molecular chain movement. The viscosity plays a crucial role in cell growth; low viscosity results in less resistance and larger cell size. In contrast, high viscosity leads to a smaller cell diameter and increased cell diameter distribution, hindering cell growth [23]. Avinash et al. increased the melt viscosity by adding calcium, which reduced the rate of gas escape, leading to a dense porous structure [24].

In the injection-molding machine, the low viscosity state of the PP blend helps mix the gas and melt it more uniformly within the barrel. After the melt is injected into the mold, it is in a state of low shear rate and high viscosity, which limits the growth of cells and results in a smaller average cell size. The early high shear rate stage has an impact on cell nucleation, while the later low shear rate stage affects cell growth, and both stages directly affect the final cell structure. The viscosity profile also partially reflects the melt strength of the blend. The lack of melt strength in MPP can lead to cell rupture, as it cannot support cell growth. Adding the right amount of SEBS can increase the melt strength of the composite, boost cell nucleation and growth, prevent premature cell rupture, and enhance the foaming effect [25].

### 3.3. Elastomer Distribution

The elastomer distribution of PP/SEBS composites is shown in Figure 6. The voids in Figure 6 result from the exfoliation of SEBS molecules that have agglomerated. As the SEBS content increases, both the number and size of the voids gradually increase and reach a maximum of 20% SEBS content. This increase in voids affects the mechanical strength of the material. However, it can be observed that the number and size of voids in the foamed samples are smaller compared to the solid samples, indicating better material dispersion in the foamed samples and the ability of cells to improve the material’s structure. 

Figure 6 shows that the composite has a typical “sea-island” structure, with SEBS particles as the dispersed phase in a regular spherical shape and the PP matrix as the continuous phase. The continuous phase generally determines the material’s fundamental properties such as modulus, strength, and elasticity. The dispersed phase significantly influences the material’s toughness, gas diffusivity, optical properties, and heat transfer in polymer alloys. The presence of SEBS particles in the composite leads to a non-uniform stress field and concentration. Under tensile stress, numerous cracks and shear bands are generated around the SEBS particles, especially near the equatorial plane, consuming significant energy and enhancing the blend’s toughness. The mutual interference of stress fields between multiple crazes and the presence of rubber phase particles can induce and control crazing, absorbing a large amount of impact energy and protecting the material from damage when impacted [26].

The distribution of cell numbers and sizes in foamed materials with varying SEBS contents was analyzed using Imagine Pro, with over 200 cells counted in each plot. Although the cell size distribution is bimodal, the average cell diameter still reflects the impact of foaming on the properties of the composite. Results showed that the average diameter of SEBS in the foamed samples decreased from 849 nm to 722.5 nm compared to solid samples with a 20% SEBS content. The smaller particle size resulted in a more homogeneous distribution of nucleation sites, which enhanced cell nucleation and increased cell density. The shift in particle size was due to the combination of cell growth and melt solidification. As the melt solidified and crystallized, the growth of cells altered the distribution of particles, resulting in a change in particle size due to the varying times required for complete solidification of the elastomer matrix. Previous research has indicated that the toughness of composites is strongly correlated with the size of elastomer particles [27]. Thus, the toughness of the matrix can be affected by modifying the size of the elastomer particles.

### 3.4. Cell Structure

Figure 7 shows the cell morphology of the composites. During the injection process, the melt in close proximity to the inner wall of the mold cools rapidly, hindering cell nucleation and resulting in smaller final cell sizes. On the other hand, the melt in the core region is at a higher temperature, enabling smooth cell nucleation and leading to larger final cell sizes. This creates a more distinct transition layer-core layer-transition layer structure. As depicted in Figure 7a, without the addition of SEBS, the number of cells is limited, and their diameter is relatively large. The cell structure was notably improved after adding 15% SEBS, when the cell size was reduced, and the shape became more regular. According to Colton and Suh [28,29], the fundamental way to enhance cell nucleation and refine cell morphology is to lower the energy barrier of cell nucleation. Introducing heterogeneous nucleating agents can decrease the energy barrier, thus reducing the resistance to cell nucleation. When 20 wt% SEBS was added, the cells’ aspect ratio and offset angle reached the minimum value of 2.3 and 12°, respectively. 

Figure 8 illustrates the curves of the cell structure parameters. It can be observed from Figure 8a,b that upon the addition of 15% SEBS, the cell diameter in the core and transition layers was reduced to 47.2 and 16.3 μm, respectively. We measured the densities of MPP and MPP/SEBS composites using thermogravimetric analysis. The density of MPP is 0.8 g/cm^3^, the density of MPP/SEBS composites is between 0.7 and 0.8 g/cm^3^. Due to the small density difference and similar expansion ratio between composite materials with different SEBS contents, the cell density was the main focus of our analysis. In contrast, the cell densities increased to 8.79 × 10^5^ and 15.6 × 10^5^ units/cm^3^ respectively. This is because the addition of SEBS effectively lowered the cell nucleation barrier, provided more heterogeneous nucleation sites, and increased the cell nucleation rate. Meanwhile, the increased viscosity of the blend due to the addition of SEBS inhibited cell growth and stabilized the cell structure, thus elevating the cell density.

### 3.5. Toughening Effect

Figure 9 shows the impact strength and impact cross-section of the composite. The addition of SEBS notably boosted the impact strength of MPP. The impact strength of the foamed MPP samples was higher than that of the solid samples, demonstrating the toughening effect of the cells on the matrix. However, as the SEBS content increased, the impact strength of the foamed samples gradually declined compared to the solid samples. The fracture surface of MPP in Figure 9a is smooth and free of noticeable plastic deformation, exhibiting brittle fracture. As the SEBS content increases, the fracture surface begins to exhibit more prominent plastic deformation characteristics. As shown in Figure 9e, there is significant tearing evident at 20 wt% of SEBS content, indicating that the matrix absorbed a significant amount of energy during the fracture process, resulting in a significant increase in toughness.

Newman et al. proposed the crazing and shear band theory to explain the toughening mechanism of elastomers [30]. Based on this theory and the results of the impact and tensile tests in this experiment, a hypothetical qualitative model is proposed (Figure 10). Figure 10 shows the crack extension schematically. A large number of SEBS particles are dispersed in the matrix, serving as stress concentration points when subjected to external forces, leading to the formation of multiple cracks and shear bands. The shear bands and cracks that form between adjacent SEBS particles interfere with each other and absorb a significant amount of energy, hindering the development of destructive cracks. The combination of cracks and shear bands enhances the composite’s toughness. 

The cells in the composite blunt the crack tip and create secondary cracks, which absorb energy through cell deformation. The cells also cause crazing and shear banding, similar to the effect of elastic particles, but the toughening effect is less pronounced than elastic particles. The main reason for this is the presence of cells, which reduces the density of PP molecular chains per unit volume, making it easier for the molecular chains to move under the influence of external forces, resulting in decreased toughness. Hence, as the SEBS content increases, the impact strength of solid samples becomes higher than that of foamed samples.

Figure 11 shows the tensile stress-strain curves of the composites. As observed, the incorporation of SEBS significantly improved the elongation of the blends, with the foamed samples reaching a breaking elongation of 825% at 20 wt% SEBS content. Despite this improvement, the breaking elongation of the foamed samples was lower than that of the solid samples due to the fact that the defective cells served as stress concentration points, reducing the effective cross-sectional bearing area and leading to premature failure. Figure 10 also shows that the addition of SEBS decreased the yield strength of the composite. PP has a higher modulus than SEBS, and MPP has a higher yield strength. As SEBS, a low-modulus material, was incorporated, the modulus of the composite decreased with increasing SEBS content, resulting in a decreased yield strength. The lower yield strength of the foamed materials compared to solid materials can be attributed to the reduced effective support area from the presence of cells and the occurrence of destructive cracking from defective cells. The mechanical properties tests revealed that while the foamed materials did not perform as well as solid materials, SEBS and cells had a synergistic effect in improving the toughness of the composites. As such, future research should prioritize enhancing the foaming effect and increasing the strength of foamed materials to develop lightweight reinforced composites that meet the actual requirements of automotive applications.

On the other hand, the elastomeric properties of SEBS can be seen to greatly enhance the toughness of the blend, as illustrated in Figure 8. Previous research has demonstrated that the fracture toughness of foam materials is affected by several factors, such as their solid strength, cell structure characteristics, and relative density [31]. The relative density of foam materials is always less than one, thus resulting in a tensile strain usually lower than that of solid materials. Therefore, the influence of various foaming process parameters on the structure and properties of composites should have been considered in the experiments. This is important as it could affect the toughening effect of the materials and requires further investigation.

## 4. Conclusions

The MPP/SEBS solid and foamed composites were prepared, and their properties were tested to investigate the toughening mechanism of SEBS on the composites. Results showed that the addition of SEBS increased the impact strength of the composites by a factor of three and increased the elongation at break by 11.2 times compared to the solid MPP material. The addition of SEBS disrupted the regular arrangement of PP molecules, refined the grain size, reduced the cell size and increased cell density. The cells in the core layer and transition layer in the vertical cross-section reached 50.7 and 29.5 μm, respectively, with cell densities of 4.48 × 10^5^ and 22.76 × 10^5^ cells/cm^3^ respectively, while the dense cells also improved the impact strength of the composites. The results showed that the addition of SEBS improved the adhesion between the phase interfaces in the blended system, refined the grain size, enhanced the foaming effect, and effectively improved the composites’ impact resistance and tensile properties. The presence of cells also had a specific toughening effect on the composites.

## Figures and Tables

**Figure 1 polymers-15-01503-f001:**
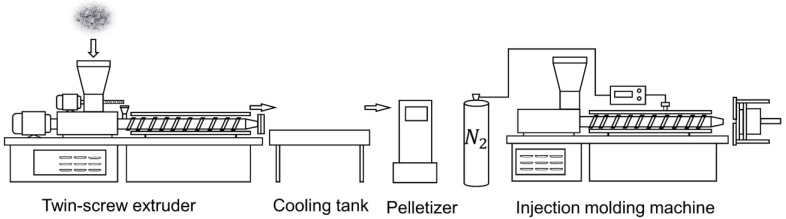
Supercritical microcellular foaming technology.

**Figure 2 polymers-15-01503-f002:**
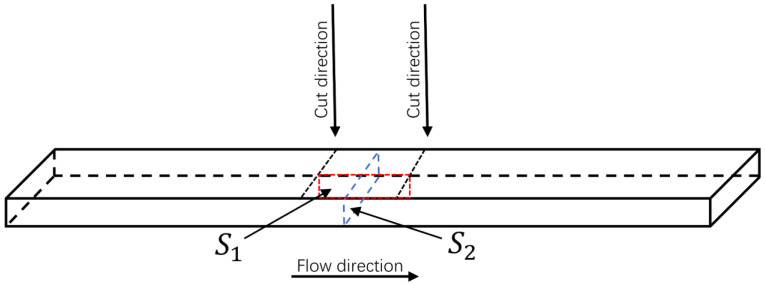
Schematic diagram of sample segmentation ($S_{1}$: parallel section, $S_{2}$: vertical section).

**Figure 3 polymers-15-01503-f003:**
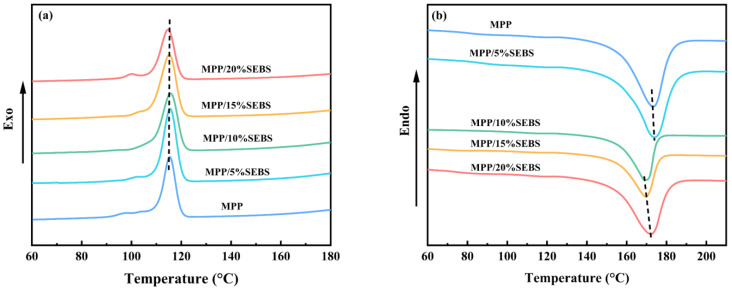
(**a**) Crystallization curve, (**b**) Melting Curve.

**Figure 4 polymers-15-01503-f004:**
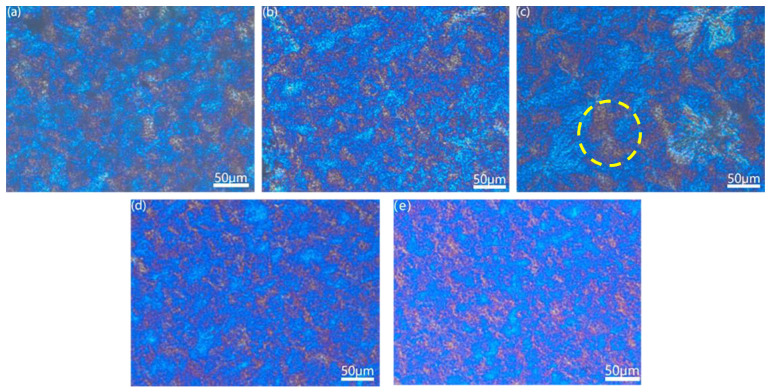
Crystal morphology under polarized optical microscope. ((**a**)—MPP; (**b**)—MPP/5%SEBS; (**c**)—MPP/10%SEBS; (**d**)—MPP/15%SEBS; (**e**)—MPP/20%SEBS).

**Figure 5 polymers-15-01503-f005:**
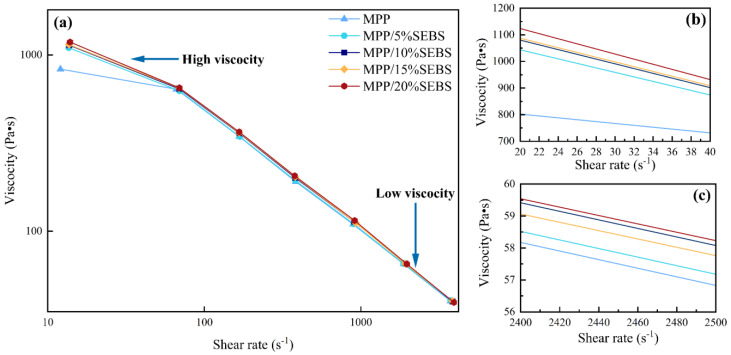
Shear viscosity curves (**a**) general plot, (**b**) partial magnification at a low shear rate, and (**c**) partial magnification at a high shear rate.

**Figure 6 polymers-15-01503-f006:**
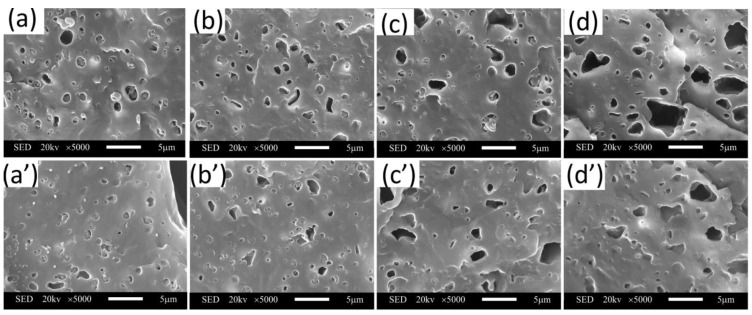
Elastomer distribution: (**a**–**d**: solid, **a’**–**d’**: foamed). (**a**,**a’**: MPP/5%SEBS); (**b**,**b’**: MPP/10%SEBS); (**c**,**c’**: MPP/15%SEBS); (**d**,**d’**: MPP/20%SEBS).

**Figure 7 polymers-15-01503-f007:**
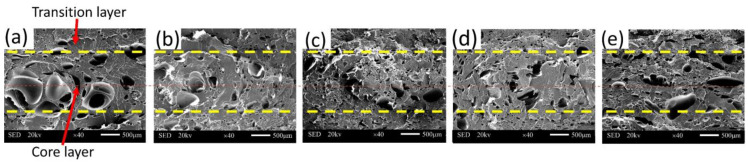
Cell structure: (**a**)—MPP, (**b**)—MPP/5%SEBS, (**c**)—MPP/10%SEBS, (**d**)—MPP/15%SEBS, (**e**)—MPP/20%SEBS.

**Figure 8 polymers-15-01503-f008:**
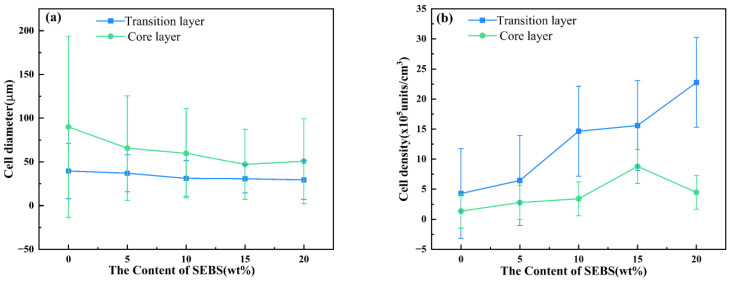
Cell structure parameters: (**a**) The cell diameter in the vertical section, (**b**) The cell density in the vertical section.

**Figure 9 polymers-15-01503-f009:**
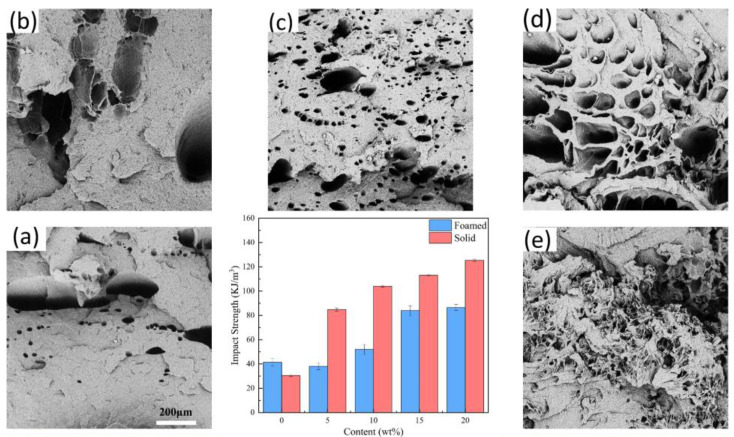
Impact profile and impact properties of foamed samples ((**a**)—MPP, (**b**)—MPP/5%SEBS, (**c**)—MPP/10%SEBS, (**d**)—MPP/15%SEBS, (**e**)—MPP/20%SEBS).

**Figure 10 polymers-15-01503-f010:**
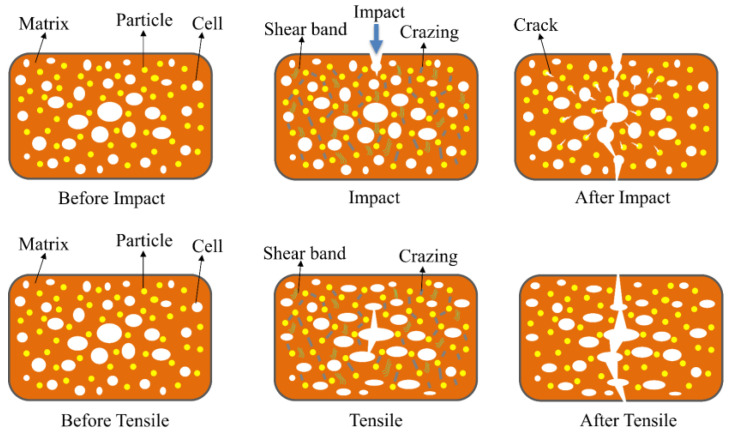
Schematic diagram of crack extension.

**Figure 11 polymers-15-01503-f011:**
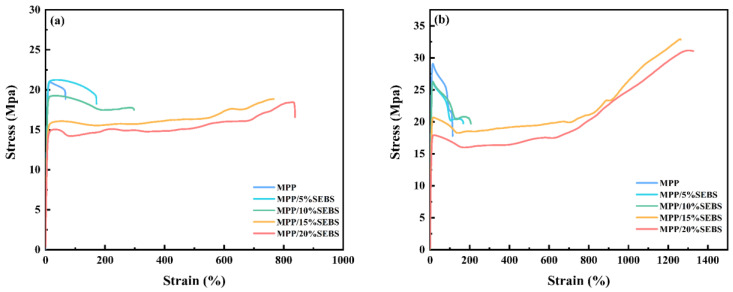
Tensile stress-strain curves of composite: ((**a**)—foamed sample; (**b**)—solid sample).

**Table 1 polymers-15-01503-t001:** Material components (wt%).

Components	PP	PP-g-MAH	SEBS
MPP	95 wt%	5 wt%	0 wt%
MPP/5%SEBS	90 wt%	5 wt%	5 wt%
MPP/10%SEBS	85 wt%	5 wt%	10 wt%
MPP/15%SEBS	80 wt%	5 wt%	15 wt%
MPP/20%SEBS	75 wt%	5 wt%	20 wt%

**Table 2 polymers-15-01503-t002:** Crystallization data of PP/SEBS composites.

Sample	$T_{C}$ (°C)	$\Delta H_{m}\left(J/g\right)$	$T_{m}$ (°C)	$X_{C}(\%)$
MPP	114.5	68	173.3	34.2
MPP/5%SEBS	116.7	73.2	177.6	38.9
MPP/10%SEBS	115.8	65	177.3	37
MPP/15%SEBS	115.3	61.9	178.6	36.5
MPP/20%SEBS	114.7	58.5	180.9	35.3

## Data Availability

The raw/processed data required to reproduce these findings cannot be shared at this time as the data also forms part of an ongoing study.

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
