# Peer review of "The Cellular Structure and Toughness of Hydrogenated Styrene-Butadiene Block Copolymer Reinforced Polypropylene Foams"

_polymers, 2023, doi:10.3390/polym15061503_

Round 1
Reviewer 1 Report (Previous Reviewer 1)
The provided revisions and additions to the resubmitted manuscript contributed to the improvement of the quality of the manuscript and made it more suitable for publication. I recommend making some additional revisions before final acceptance of the paper.
1. In addition to the confidence intervals in Fig. 8, a, they should also be shown in Fig. 8, b (the cell densities in the transition and core layers against the SEBS content).
2. The additional corrections of English writing throughout the text will be good. As an example: caption to Fig. 8; “The cell diameter of vertical section”; it seems this should be as “The cell diameter in the vertical section”, and so on.
3. The data on the average cell size and their volume concentration (Fig. 8) should be compared to the corresponding expansion factors for the synthesized composites. Accordingly, a discussion regarding this point is desirable. In particular, the rough estimates of the specific volume of voids (the total volume of voids per cubic centimeter) using the presented data give the values no more than a few hundredths. This corresponds to very low values of the expansion factor. Is it real?
Author Response
Please see the attachment.

Reviewer 2 Report (New Reviewer)
Well done paper. The blending of foamed PP with SEBS is an interesting idea to improve impact strength. The methods are well described and according to the state of the art. The authors did not make only mechanical testing, they have done the necessary work in microscopy, too. So we learn about the whole approach Material - Structure - Properties.
Please look to the file in which I made some remarks.

Author Response
Please see the attachment.

Reviewer 3 Report (New Reviewer)
As preparation of materials possessing the desired properties is an area continuing to arouse enthusiasm throughout the world, research dedicated to the improvement of the mechanical features of polymers definitely deserves attention. The article presents a good description of the PP/SEBS composites supplying the experimental results with clarifying comments and explanations. The manuscript may definitely be recommended for publication after making the corrections listed below.
Corrections to be made
1. The meaning of “β-NA” remains unclear. (page 2)
2. The phrase “at a constant temperature” is repeated in the sentence “The samples were held at a constant temperature at a constant temperature of 230°C for 5 min after completion“. (page 3)
3. Use the multiple form of “equation” in the sentence “The remaining parameters were calculated separately using the following equation [19,20]”. (page 4)
4. “Therefor” instead of “therefore”. (page 7)
5. Compose sentences in either present or past tense. Avoid using the tenses alternatingly as in “As the melt solidifies and crystallized, the cell growth altered the particle distribution, leading to a change in particle size due to the varying times required for complete solidification of the elastomer matrix.”
6. It would be better if the authors make it clear what the yellow lines stand for in Figure 7.
Author Response
Please see the attachment.

This manuscript is a resubmission of an earlier submission. The following is a list of the peer review reports and author responses from that submission.
Round 1
Reviewer 1 Report
The manuscript “The cellular structure and toughness of hydrogenated styrene-2 butadiene block copolymers reinforced polypropylene foams” by Wei Guo et al. is devoted to the study of the effect of styrene-ethylene-butadiene-styrene (SEBS) copolymer adding on the cellular structure and toughness of SEBS/polypropylene composites. The authors applied a number of test methods (the differential scanning calorimetry, rheometry, scanning electron microscopy, polarization microscopy, mechanical tests) to characterize this effect. In general, the presented results are of particular interest to the “Polymers” readership specialized in the synthesis and characterization of polymer-based functional materials. However, the style of manuscript presentation is hard-to-follow and needs major revisions to make it acceptable for publication. The corresponding comments are listed below.
1. The quality of scientific English writing is confusing. There are questionable definitions and terms throughout the text. For example, “The multiple crazes theory” (line 83) - what does it mean? What theory of crack formation do the authors have in mind? Another example, the text fragment in line 357,
“At the same time, the microcellular also had specific….”. Following English grammar and wording rules, “microcellular” is the adjective. But where is the substantive? In this regard, I recommend that the authors undertake careful editing of English scientific writing, perhaps with the help of colleagues with relevant experience.
2. Again about “the multiple crazes theory” (line 83). I did not find in the manuscript any theoretical consideration of the crazes formation in the usual sense of this term. In my opinion, any theory of this type should be based on reliable quantitative relationships between the analyzed parameters and allow predicting the behavior of the system depending on these parameters. The related text fragment (lines 303-319) and supporting Fig. 10 cannot be considered as a theory, but only as hypothetical qualitative model. The respective claims of the authors should be substantially limited.
3. Values of cell diameters given in the Abstract with an accuracy of 5 digits (line 24, 156.98 micrometers and 66.7 micrometers) are very confusing. Only one question arises: did the authors use any statistical processing of SEM data or did they randomly take the first numbers that came across from the samples? And again about the English grammar, it should be the “cell diameter”, not the “cellular diameter”.
4. Please, provide the same notations for the crystallinity percentage in Eq. 1 and in Table 2. It is necessary to give the determination errors (in particular, confidence intervals) for the values presented in Table 2. In some cases, the presented data look confusing in a statistical sense (for example, compare the presented enthalpy values for 10% SEBS (65 J/g, 2 significant digits) and for 20% SEBS (58.45 J/g, 4 significant digits). The reported data should be characterized by approximately the same accuracy of determination.
5. Discussion of the results of polarization microscopy presented in Figs. 4 should be made more extended and clear for readers who are not experienced in the application of this method. Also, “the crystalline-linearly values” - what does this definition mean?
6. Line 254, “from 849.03 nm to 722.54 nm” - such accuracy seems unrealistic. Regarding Fig. 6, it seems that there are no differences in the structure of the foamed (a’-d’) and solid (a-d) samples. Additional comments regarding this point are desirable.
7. What is a reason to use the additional supercritical fluidic foaming of the synthesized composites? As follows from Fig. 11, the “solid” samples exhibit better mechanical properties compared to foamed samples (yellow and pink lines). Additional comments regarding this point are desirable.
Reviewer 2 Report
The current manuscript by Guo et al. presents a technological scheme for the optimization of a material’s strength by varying the concentration of a new additive (styrene-ethylene-butadiene-styrene copolymers) to polypropylene materials. The manuscript shows how different concentrations of the additive change the viscosity during processing and the materials' impact and tensile strengths. As a result, the authors observe an increase in the mechanical properties several times, similar to other authors that performed similar work, as cited by the authors themselves (see refs. [11-15] and the introduction).
As mentioned above, the study appears to be unoriginal. It does not improve the current state-of-the-art, nor it provides a novel insight into any of the technological processes involved along the materials characterization. The discussion of the effects within the manuscript appears to be vague and unrelated to each other – for instance, it is unclear why the authors characterize the viscosity of the fluid and discuss the foam generation if the macroscopic density of the materials and its effect on the mechanical strength are not discussed at all.
Many of the presented results are not clear – there are “transitional layers” that are not defined, pore sizes that are bimodal but are averaged, materials with arbitrary densities are compared mechanically (cell number density is different from macroscopic density), some Figures are ill-defined (how do we know that the defects in figure 9 come from breakage but not the structure of your material, the captions in the Figure itself are missing).
As a result of the above concerns, I recommend the manuscript is rejected from publication in its current form.
